# The Immune Signature of CSF in Multiple Sclerosis with and without Oligoclonal Bands: A Machine Learning Approach to Proximity Extension Assay Analysis

**DOI:** 10.3390/ijms25010139

**Published:** 2023-12-21

**Authors:** Lorenzo Gaetani, Giovanni Bellomo, Elena Di Sabatino, Silvia Sperandei, Andrea Mancini, Kaj Blennow, Henrik Zetterberg, Lucilla Parnetti, Massimiliano Di Filippo

**Affiliations:** 1Section of Neurology, Department of Medicine and Surgery, University of Perugia, 06132 Perugia, Italy; lorenzo.gaetani@unipg.it (L.G.);; 2Institute of Neuroscience and Physiology, Department of Psychiatry and Neurochemistry, The Sahlgrenska Academy at the University of Gothenburg, 431 41 Mölndal, Sweden; 3Clinical Neurochemistry Laboratory, Sahlgrenska University Hospital, 431 41 Mölndal, Sweden; 4Department of Neurodegenerative Disease, UCL Institute of Neurology, Queen Square, London WC1N 3BG, UK; 5UK Dementia Research Institute at UCL, London WC1E 6BT, UK; 6Hong Kong Center for Neurodegenerative Diseases, Clear Water Bay, Hong Kong 518172, China; 7Wisconsin Alzheimer’s Disease Research Center, University of Wisconsin School of Medicine and Public Health, University of Wisconsin-Madison, Madison, WI 53792, USA

**Keywords:** multiple sclerosis, cerebrospinal fluid, biomarkers, oligoclonal bands, proximity extension assay, machine learning

## Abstract

Early diagnosis of multiple sclerosis (MS) relies on clinical evaluation, magnetic resonance imaging (MRI), and cerebrospinal fluid (CSF) analysis. Reliable biomarkers are needed to differentiate MS from other neurological conditions and to define the underlying pathogenesis. This study aimed to comprehensively profile immune activation biomarkers in the CSF of individuals with MS and explore distinct signatures between MS with and without oligoclonal bands (OCB). A total of 118 subjects, including relapsing–remitting MS with OCB (MS OCB+) (*n* = 58), without OCB (MS OCB−) (*n* = 24), and controls with other neurological diseases (OND) (*n* = 36), were included. CSF samples were analyzed by means of proximity extension assay (PEA) for quantifying 92 immune-related proteins. Neurofilament light chain (NfL), a marker of axonal damage, was also measured. Machine learning techniques were employed to identify biomarker panels differentiating MS with and without OCB from controls. Analyses were performed by splitting the cohort into a training and a validation set. CSF CD5 and IL-12B exhibited the highest discriminatory power in differentiating MS from controls. CSF MIP-1-alpha, CD5, CXCL10, CCL23 and CXCL9 were positively correlated with NfL. Multivariate models were developed to distinguish MS OCB+ and MS OCB− from controls. The model for MS OCB+ included IL-12B, CD5, CX3CL1, FGF-19, CST5, MCP-1 (91% sensitivity and 94% specificity in the training set, 81% sensitivity, and 94% specificity in the validation set). The model for MS OCB− included CX3CL1, CD5, NfL, CCL4 and OPG (87% sensitivity and 80% specificity in the training set, 56% sensitivity and 48% specificity in the validation set). Comprehensive immune profiling of CSF biomarkers in MS revealed distinct pathophysiological signatures associated with OCB status. The identified biomarker panels, enriched in T cell activation markers and immune mediators, hold promise for improved diagnostic accuracy and insights into MS pathogenesis.

## 1. Introduction

Multiple sclerosis (MS) is an autoimmune inflammatory disorder that affects the central nervous system (CNS), leading to damage in the myelin, neurons and axons [1]. The diagnosis of MS in its early stages is achieved through continually updated diagnostic criteria, relying on the integration of multiple investigations as no single diagnostic test is available [2,3]. The primary diagnostic steps involve assessing the clinical presentation and spatial–temporal occurrence of inflammatory lesions observed via magnetic resonance imaging (MRI) of the CNS. Additionally, cerebrospinal fluid (CSF) examination provides key information for differential diagnoses and offers evidence of intrathecal IgG synthesis [2]. The positivity of biomarkers indicating intrathecal IgG synthesis holds significant prognostic value in MS, signaling an increased risk of disease recurrence even after the initial clinical manifestation [4]. Intrathecal IgG synthesis can be detected by using isoelectrofocusing (IEF) to assess oligoclonal bands (OCB) in paired CSF and serum samples, with OCB presence in CSF but not serum indicating intrathecal B-cell activity [5]. Nevertheless, OCB lack the ability to offer a quantitative measure, and they solely reflect intrathecal B-cell activity, thus not providing insights into the complex immunological milieu that could characterize the CSF in a chronic inflammatory condition like MS [6]. Moreover, 5–10% of individuals with relapsing–remitting MS (RRMS) do not exhibit OCB, and this percentage tends to be higher during the initial phases of the disease and in Eastern countries [7]. This subset of patients presents a potential challenge in terms of diagnosis [8] and prompts inquiries into whether MS cases with and without OCB might possess distinct immune pathogeneses. A more precise biomarker-based characterization of MS is, therefore, needed. 

CSF, owing to its proximity to the CNS, offers an advantageous focus for biomarker exploration. The identification of MS-related pathophysiological biomarkers has proven challenging due, in part, to sensitivity issues. While multiplex analyses have been conducted to characterize the immune signature of CSF in people with MS (pwMS), the results have exhibited heterogeneity across studies, partly due to the limited scope of measured biomarkers, primarily focused on a small subset of inflammatory proteins, and to the challenges posed by antibody cross-reactivity and inter-assay variability [9]. In recent years, targeted quantitative proteomics has emerged as a promising tool for developing disease-specific protein signatures as biomarkers, furthering the understanding of the molecular mechanisms underlying neurological disorders [10]. The application of proximity extension assay (PEA) technology, combining antibody-based binding with DNA-based signal amplification [11,12], has shown considerable potential for quantitatively measuring protein levels in various body fluids, including CSF, in neurological diseases [10,13,14].

In the present study, we utilized PEA technology to comprehensively measure a panel of biomarkers reflecting immune activation in CSF samples obtained from pwMS with and without OCB and a control group. Additionally, we quantified CSF levels of neurofilament light chain (NfL), a well-established biomarker of axonal damage [15], in order to identify the prevailing immunological profile more closely linked with axonal injury. By applying robust machine-learning statistical models, the aim of the study was to identify combinations of biomarkers able to differentiate pwMS with and without OCB from controls, as well as to provide novel insights into the pathophysiology of the disease.

## 2. Results

### 2.1. Correlation Analysis of CSF PEA-Tested Proteins and NfL

Out of the 92 proteins determined through the PEA technique, 47 had less than 15% missing values (values above the lower limit of detection) in both controls with other neurological diseases (OND) and MS. From the correlation and cluster analyses, it emerged that among the measured proteins, some of them strongly correlated with each other in pwMS. The biggest cluster (cluster 1) consisted of TRAIL, IL-10RB, uPA, HGF, CX3CL1, Beta-NGF, OPG, DNER, ADA, SCF, SIRT2, TGF-alpha, CSF-1, PD-L1, Flt3L, FGF-5, CD40, CDCP1, TWEAK, VEGF-A and LIF-R. Another cluster of highly correlated proteins (cluster 2) consisted of IL-12B, CD5, TNFRSF9, CXCL9, TNFB, CXCL10 and CXCL11 (Figure 1). Evidence of interaction is apparent for most proteins within the two clusters. However, in cluster 1, DNER, SIRT2, LIF-R, IL-10RB and CDCP1 were previously not found to be associated with the other proteins of cluster 1. Similarly, within cluster 2, CD5 was not known to interact with the other proteins (Figure 2). For abbreviations of PEA-tested proteins, see Appendix A. Among the measured PEA-tested biomarkers within the RRMS group, five of them showed a weak positive but significant correlation with CSF NfL (Figure 3). 

### 2.2. Biomarkers Efficacy in Discriminating between RRMS and OND

We applied logistic regression by assuming age and sex as covariates to assess which of the inflammatory proteins was considered more altered in the whole RRMS cohort vs. OND comparison (Figure 4). The proteins showing a q-value (*p*-value adjusted for age, sex and FDR) < 0.05 were: CD5 (AUC: 0.87, 95% CI 0.80–0.94; q = 0.002), IL-12B (AUC: 0.81, 95% CI 0.73–0.89; q = 0.007), TNFB (AUC: 0.78, 95% CI 0.68–0.86; q = 0.01), TNFSF14 (AUC: 0.70, 95% CI 0.60–0.80; q = 0.01), TNFRSF9 (AUC: 0.65, 95% CI 0.54–0.76; q = 0.04) and MIP-1-alpha (AUC: 0.59, 95% CI 0.48–0.70; q = 0.02) (Figure 4 and Appendix A). We then applied a multinomial LASSO regression to differentiate between MS OCB+/− and OND, which resulted in logistic models considering CD5, IL-12B, TNFB, MCP-1, CXCL1, CXCL9 and NfL. Due to the poor results obtained in classifying MS OCB− (54% OCB− identified as MS and 46% identified as OND) (Table 1), we performed separate LASSO analyses for MS OCB+ vs. OND and MS OCB− vs. OND.

### 2.3. Multivariate Analysis of PEA-Tested Proteins in MS OCB+ vs. OND

We created an age-matched training set composed of 18 OND and 22 MS OCB+ by pairing as much as possible the age histograms of the two groups (*p*-value age = 0.73 with t-test after matching). By cross-validation, we identified an optimal penalization parameter λ of 0.0501 corresponding to the following coefficients for z-scored biomarkers: intercept (0.61), IL-12B (1.14), CD5 (0.94), CX3CL1 (−0.36), FGF-19 (−0.27), CST5 (0.23), MCP-1 (−0.16). Coefficients relative to not z-scored NPX values are reported in Appendix A. In the training set, the logistic model had a sensitivity of 91% and a specificity of 94% in detecting MS OCB+ vs. OND. The age-unmatched OND (*n* = 18) and MS-OCB+ (*n* = 36) groups were then used as validation sets. In this set, the model had a sensitivity of 81% and a specificity of 94% (Table 2).

### 2.4. Multivariate Analysis of PEA-Tested Proteins in MS OCB− vs. OND

As for the previous comparison, we created an age-matched training set composed of 15 OND and 15 MS OCB− by pairing the age histograms of the two groups as much as possible (*p*-value age = 0.51, *t*-test). The LASSO regression identified an optimal penalization parameter λ of 0.0945 and the following coefficients for z-scored biomarkers: intercept (0.094), CX3CL1 (−0.58), CD5 (0.41), NfL (0.20), CCL4 (0.10) and OPG (−0.06). Coefficients relative to not z-scored NPX values are reported in Appendix A. In the training set, the model had a sensitivity of 87% and a specificity of 80% in detecting MS OCB− vs. OND. The age-unmatched OND (*n* = 21) and MS OCB+ (*n* = 9) were used as a small validation set. In this set, the model had a sensitivity of 56% and a specificity of 48% (Table 3).

## 3. Discussion

From the univariate analysis of PEA data, we identified a panel of six CSF immunological proteins with the highest discriminatory power in the comparison between MS and controls. This panel was made of CD5, IL-12B, TNFB, TNFSF14, TNFRSF9 and MIP-1-alpha. Most of these markers are involved in T cell activation. CD5 (cluster of differentiation 5) is a T cell surface glycoprotein that may act as a receptor in regulating T cell proliferation [16]. IL-12B (interleukin 12 subunit beta), a component of the IL-12 cytokine, plays a critical role in promoting the differentiation of T cells into T helper 1 (Th1) cells, and it has a clear involvement in MS, being a genetic risk factor for the disease [17]. TNFRSF9 (tumor necrosis factor receptor superfamily member 9) is primarily expressed in antigen-presenting cells such as B cells, macrophages and dendritic cells, and it promotes T-cell activation and regulates proliferation and survival of T cells [18]. A soluble form of TNFRSF9 released by activated lymphocytes has already been found to be significantly high in CSF and serum of pwMS with clinically active disease [19], suggesting a potential of this protein as a marker for MS. MIP-1-alpha (macrophage inflammatory protein-1 alpha) are chemokine recruiting and activating immune cells, particularly monocytes, macrophages and T cells, to areas of inflammation, including MS brain lesions [20]. TNFSF14 (tumor necrosis factor ligand superfamily member 14) is a glycoprotein involved in dendritic cell maturation, and in its soluble form, it may act as an inhibitor of T-cell activation [21]. In MS serum, an increase of this marker has already been documented during disease activity, suggesting that soluble TNFSF14 is protective and may act to limit inflammation [22]. Of interest, a polymorphism of the gene coding TNFSF14 has been shown to increase the risk of MS in a genome-wide association study [23], with allelic variants of the gene being involved in the risk of MS in the Italian population as well [24].

Among the markers included in the panel, TNFB (tumor necrosis factor beta) stands out as a particularly noteworthy one that has recently gained significant attention in the context of MS. TNFB plays a crucial role in the maintenance of the immune system and is known to be involved in cellular cytotoxicity, lymphoid neogenesis and the formation of tertiary lymphoid-like structures (TLS) [25]. The presence of TLS in the subarachnoid space has been linked, in MS, to neuronal loss in the cortical grey matter, which, in turn, is a risk factor for faster disability progression [26]. Recent studies have reported higher levels of TNFB in the CSF of individuals with RRMS exhibiting a high number of cortical lesions compared to RRMS patients with a lower burden of cortical involvement [27]. Additionally, CSF TNFB levels have been found to be increased in people with progressive MS who showed a high burden of grey matter demyelination and immune cell infiltration in post-mortem brains. Furthermore, elevated TNFB mRNA levels have been observed in the meninges of pwMS with secondary progressive MS, which aligns with increased CSF TNFB levels [27]. Our findings reinforce the evidence of CSF TNFB as a biomarker in MS with relevant pathophysiological meaning. Further research in this area could offer insights into the underlying mechanisms and potential therapeutic targets of MS.

To mitigate the risk of data overfitting, a common concern when dealing with multiplex analyses that yield numerous outcome variables, we employed a machine learning methodology grounded in penalized regression analysis, i.e., the LASSO. The LASSO holds the potential to uncover the most compact set of innovative markers, ensuring both heightened sensitivity and specificity in distinguishing individuals with MS from the control group. By applying multinominal LASSO, we identified a set of proteins consisting of biomarkers of T cell activation (CD5, IL-12B) and chronic meningeal inflammation (TNFB) already discussed, together with chemokines involved in T cell migration (CXCL1 and CXCL9) and monocytes and memory T cell migration (MCP-1), and a biomarker of axonal damage (NfL). Of note, it is intriguing to observe that this model exhibited contrasting performances in pwMS based on the presence or absence of OCB. 

The model demonstrated notable efficacy in classifying MS OCB+ (85% accuracy), yet its precision was diminished in the case of MS OCB−. For the latter group, merely 54% of patients were correctly identified as having MS, with the remaining 46% inaccurately labeled as controls. For this reason, we applied binomial LASSO to classify MS OCB+ and MS OCB− separately from controls. In this analysis, we found that the best model to identify MS OCB+ was made of IL-12B, CD5, CX3CL1, FGF-19, CST5 and MCP-1. Among these, fractalkine (CX3CL1) is another protein reflecting the activation of T cells, similar to IL-12B and CD5. CX3CL1 has been shown to increase IFN-γ and TNF-α gene expression and IFN-γ secretion by CD4(+) T cells derived from RRMS patients [28]. Further, in people with RRMS, CSF levels of CX3CL1 have already been demonstrated to be increased [28]. Cystatin-D (CST5) has shown potential as a relapse marker in an independent cohort of pwMS [14]. Furthermore, it has been found to be increased in the CSF of patients with severe traumatic brain injury [29]. Of note, in both these studies, CST5 was measured by means of PEA. CST5 is an inhibitor of lysosomal and secreted cysteine proteases, originally purified from saliva, with an undefined role [30]. The association between CSF CST5 relapses in pwMS and traumatic brain injury suggests that this marker might serve as a proxy for neuronal injury, but the interpretation of its pathophysiological meaning deserves further investigation. Fibroblast growth factor 19 (FGF-19) belongs to the endocrine subfamily of fibroblast growth factors (FGFs) [31]. FGFs are ubiquitously expressed throughout the CNS on all cell types, and FGF signaling may regulate inflammation and myelination in MS since an abundance of FGF members has been documented in focal inflammatory lesions in MS [31]. However, the specific role of FGF19 in MS is yet unexplored. 

As for MS OCB−, the most accurate diagnostic model included CX3CL1, CD5, NfL, CCL4 and OPG. Therefore, also in this subgroup of pwMS biomarkers of T cell activation, CX3CL1 and CD5 participate in identifying the disease, together with a biomarker of neuronal injury (NfL), a chemokine (CCL4) and OPG. Of interest, OPG (osteoprotegerin) has been shown to suppress the mRNA expression of CCL20, a chemokine involved in Th17 cell recruitment with anti-inflammatory effects [32]. The signaling that involves OPG has been hypothesized to act as a neuroprotectant after brain damage [33]. The presence of OPG in the multivariate model distinguishing MS OCB− from controls and its absence in the model for MS OCB+ is intriguing and deserves future investigation. 

Notably, the model used to identify MS OCB+ demonstrates a satisfactory level of accuracy (91% sensitivity and 94% specificity in the training set, 81% sensitivity and 94% specificity in the validation set). The high specificity of this model is particularly relevant for its potential application in future MS diagnosis. While the sensitivity might not be excellent in the age-unmatched validation set, it is not a major concern, given the availability of other sensitive diagnostic tools, especially in terms of MRI and MRI-based diagnostic criteria. What is needed to improve the accuracy in the diagnosis of MS is the identification of a biomarker or combination of biomarkers with high specificity for the disease. The model we found, however, should be tested against more relevant clinical comparisons, particularly diseases that mimic MS, to assess its robustness. 

On the other hand, the model used for the identification of MS OCB− was found to be less effective. This suggests that MS OCB+ and MS OCB− may differ from a pathophysiological perspective, with MS OCB− showing less definite signs of immune activation in CSF. This underscores the complexity of diagnosing MS when CSF OCB are absent. As an example, in discriminating between MS and white matter lesions associated with migraine and vascular lesions, the absence of CSF OCB is the most robust independent predictor of a non-MS diagnosis [34]. Therefore, the investigation of biomarkers able to identify MS OCB− is highly relevant from a clinical perspective and warrants further research.

One of the strengths of our study relies on having coupled a thorough immunological characterization with the measurement of a well-established marker that summarizes the overall CNS pathology occurring in MS, namely CSF NfL [15]. NfL has indeed been shown to reflect both the acute neuronal injury following new focal lesion appearance, as well as the chronic neuronal damage taking place in those phases of the disease where neurodegeneration prevails on acute focal inflammation [35]. 

We found that five of the inflammation-related biomarkers in MS correlate with CSF NfL, namely MIP-1-alpha, CD5, CXCL10, CCL23 and CXCL9. All these proteins are implicated in T cell activation or migration [16,20,36], thus suggesting that CSF may provide a robust proxy for the deleterious role of intrathecal T cell activation on neuronal survival. Many of these proteins have already been found to be upregulated in the CSF and blood of pwMS and to decrease after the start of a disease-modifying treatment [14]. Our results, by showing an association with NfL, provide further evidence that these T cell-related markers reflect the pathophysiology of the disease and correlate with the severity of axonal injury in MS.

Finally, when looking at the correlations among PEA-tested proteins, we found two clusters of inflammatory markers strongly correlated with most of these proteins already known to interact with each other. In cluster 1, TRAIL, uPA, HGF, CX3CL1, Beta-NGF, OPG, SCF, TGF-alpha, CSF-1, PD-L1, Flt3L, FGF-5, CD40, TWEAK and VEGF-A have already shown evidence of interaction. As an example, most of these proteins (TRAIL, HGF, CX3CL1, Beta-NGF, SCF, TGF-alpha, CSF-1, PD-L1, Flt3L, FGF-5, TWEAK and VEGF-A) can influence the expression or activate uPA (urokinase-type plasminogen activator), which plays a pivotal role in various physiological and pathological processes, including extracellular matrix remodeling, cell migration and invasion, angiogenesis and immune response and inflammation [37].

Interesting suggestions came from the presence in the clusters of proteins not known to interact with the others, such as ADA, DNER, SIRT2, CDCP1, LIFR and IL-10 in cluster 1 and CD5 in cluster 2. DNER is the delta and notch-like epidermal growth factor-related receptor; it activates the NOTCH1 pathway, and it is known to inhibit the proliferation of neural progenitors or induce neuronal and glial differentiation during brain development [38]. Its correlation in the CSF with immune mediators suggests it may play a role in the interaction between neuronal, glial and immune cells, especially during an inflammatory chronic disease such as MS. DNER can be the target of immune activation within autoimmune encephalitis with paraneoplastic etiology, such as cerebellar degeneration in Hodgkin lymphoma [39]. The pathophysiological basis of the correlation between DNER and T-cell and B-cell markers in MS must be demonstrated. 

SIRT2 (sirtuin 2) is a protein deacetylase highly expressed in the mammalian CNS, mainly found in the cytoplasm of oligodendrocytes [40]. It is particularly expressed in the cortex, striatum, hippocampus and spinal cord, but its functions are still largely unknown [40]. Elevated levels of SIRT2 have been found in the CSF of a neurodegenerative disease such as Alzheimer’s disease, indicating its potential as a biomarker reflecting CNS damage [13]. The correlation of CSF SIRT2 with inflammatory markers in MS suggests that it might track neuronal damage secondary to immune dysfunction, but the plausibility of this association deserves further investigation. 

In the second minor cluster, different proteins belonging to chemokines were functionally correlated, including CXCL9, CXCL10 and CXCL11, together with markers of T cell activity, such as IL-12B. Some of these markers, particularly IL-12B, have clear involvement in MS and are genetic risk factors for the disease [17]. Within this cluster, CD5 has not previously shown an association with the other markers. CD5 is a T cell surface glycoprotein that may act as a receptor in regulating T cell proliferation [16]; therefore, an association with different chemokines and with markers of T cell activation in the CSF of MS can be expected. 

In conclusion, one potential limitation of our study is its retrospective design. The inclusion of participants was based on the availability of stored CSF samples over a three-year period, which may not constitute a random selection. However, we believe this does not introduce selection bias to our cohort. Our research has several strengths: (i) we addressed the clinical challenge of diagnosing MS by focusing on pwMS with and without OCB, a subset that poses diagnostic difficulties [8]. This reflects the study’s relevance to real-world diagnostic scenarios. (ii) We employed an innovative technology, i.e. PEA technology, which offers a promising tool for quantitatively measuring protein levels in various body fluids, including CSF. This innovative approach enhances the precision of biomarker exploration. (iii) To interpret the complexity of data generated by PEA, we applied a machine learning statistical approach that enhances the robustness of our findings [41], and we built training and validation sets to test the accuracy of biomarkers in discriminating between groups.

## 4. Materials and Methods

### 4.1. CSF Sampling

We selected 118 consecutive patients for this study whose CSF samples were stored in the Laboratory of Clinical Neurochemistry, Department of Medicine and Surgery, University of Perugia (Perugia, Italy). CSF samples were collected over a 3-year period (January 2014–January 2017) via lumbar puncture at the Section of Neurology, Perugia University Hospital, Perugia (Italy), using the same standard operating procedures throughout the study, as recommended [42]. Specifically, lumbar puncture was performed between 8:00 and 10:00 a.m., and CSF was collected in sterile polypropylene tubes, centrifuged for 10 min at 2000× *g*, divided into 0.5 mL aliquots and immediately frozen at −80 °C, together with serum 0.5 mL aliquots, pending analysis. After lumbar puncture, patients’ demographic and clinical data were collected in an online electronic database. The selected CSF samples came from two groups of patients who were diagnosed at the time of CSF sampling, as follows: (i) RRMS (*n*: 82), and (ii) OND (control group, *n*: 36). For all patients, CSF was collected as part of their usual diagnostic work- up. The local Ethics Committee approved the study conduction (# 1287/08 and #3933/21).

### 4.2. Selection of CSF Samples

For the MS group, we selected CSF samples from pwMS satisfying, at the time of lumbar puncture, the following inclusion criteria: (i) a diagnosis of RRMS according to the 2017 revision of the McDonald criteria [2], retrospectively applied; (ii) age between 18 and 60 years; (iii) no history of exposure, in the 30 days prior to CSF collection, to immunosuppressant or immunomodulatory therapies. For the OND group, we selected patients who underwent CSF analysis for diagnostic reasons, with the following inclusion criteria: (i) a final diagnosis of minor non-inflammatory neurological diseases; (ii) age between 18 and 60 years; (iii) no history of exposure to immunosuppressant therapies in the 30 days preceding lumbar puncture. A senior neurologist with experience in the field of MS examined all the pwMS included for this study and scored the Kurtzke’s Expanded Disability Status Scale (EDSS) [43].

### 4.3. Standard CSF Analysis

OCB pattern detection was achieved by running both serum and coupled CSF samples by means of IEF [44] on a semi-automated agarose electrophoresis system (Sebia Hydrasys, Lisses, France) followed by immunofixation with a peroxidase-labeled anti-IgG (Hydragel 9 CSF Isoelectrofocusing; Sebia, Lisses, France). An aliquot of each serum sample was appropriately diluted to adjust the IgG concentration to the same level as found in the CSF, as specified by the manufacturer. Following OCB pattern evaluations, pwMS were divided into two groups: (i) OCB negative (IEF patterns 1, 4 and 5) and (ii) OCB positive (IEF patterns 2 and 3) when ≥2 additional OCB were detected in CSF compared to serum samples. Since, in all cases, CSF was collected during the diagnostic assessment, none of the pwMS was on disease-modifying therapy at the time of lumbar puncture. 

### 4.4. CSF NfL Measurement

CSF NfL was measured in the Institute of Neuroscience and Physiology, Department of Psychiatry and Neurochemistry at the Sahlgrenska Academy, University of Gothenburg (Mölndal, Sweden), through an in-house ELISA, as already described [45]. All samples were analyzed by board-certified laboratory technicians, all blinded to clinical data, by using one batch of reagents at a time.

### 4.5. CSF PEA Testing

Inflammatory protein panel testing was performed in 2017 using the multiplex PEA technology as previously described by Olink (Uppsala, Sweden). All the samples were run on the inflammation panel, which consists of 92 biomarkers (Appendix A), with up to 96 samples tested simultaneously on each run. The Olink panel validation data are freely available online (https://www.olink.com/data-you-can-trust/validation/, accessed on 23 July 2021). The resulting data for each biomarker were expressed as normalized protein expression (NPX) values. NPX is an arbitrary unit on a log2 scale that is obtained by normalizing the concentration values to minimize inter- and intra-assay variations. A high NPX value corresponds to a high protein concentration and can be linearized by using the formula 2NPX. NPX values were subsequently z-scored to allow for a better comparison in multivariate analysis.

### 4.6. Characteristics of the Patients

The MS group was composed of 82 subjects (F/M 58/24) with a mean age of 37.9 ± 10.6 years. Clinical characteristics are summarized in Table 4. CSF IgG OCB were found in 58 people with RRMS (MS OCB+: 58/82, 70.7%). The remaining 24 people with RRMS (29.3%) did not show CSF IgG OCB (MS OCB−). No significant clinical differences were found between MS OCB+ and MS OCB− (Table 1). The OND group was composed of 36 individuals (F/M 18/18) with a mean age of 57.9 ± 16.6 years. This group included patients with headache (*n* = 16), psychiatric disorders (*n* = 13), mononeuropathy (*n* = 4) and dysmetabolic polyneuropathy (*n* = 3). RRMS and OND had significantly different mean age (*p* < 0.001) and sex distributions (*p* < 0.05). Age and sex were, therefore, considered covariates in the subsequent analyses. CSF NfL was found to be significantly higher in RRMS compared to OND (663 pg/mL (IQR: 748) vs. 361 pg/mL (IQR: 504), *p* < 0.05) (Table 4).

### 4.7. Statistical Analysis

All the analyses reported in this manuscript were performed with R-4.3.1. Descriptive statistics, such as mean and standard deviation, were used to summarize the characteristics of the patients, including age, sex distribution and number of subjects in each group (RRMS patients and OND). Fisher’s exact test was used to assess whether there were significant differences in sex distribution between RRMS patients and OND. The Mann–Whitney U-test was employed to compare the age of RRMS patients with that of OND subjects. Out of the 92 tested proteins, 45 had a call rate <85% (i.e., <85% of the samples had a valid measurement of that protein) and were therefore removed from further analysis (Appendix A). Spearman’s correlation was used to analyze the correlations between the PEA-tested proteins. Hierarchical clustering was used for ordering proteins by using correlation coefficients as distances, which were grouped according to Ward’s linkage criterion. Logistic regression was applied to assess the differential abundance of inflammatory proteins in RRMS patients compared to OND subjects and to determine the age-, sex- and false discovery rate (FDR)-adjusted *p*-values (q-values) for each protein. Area under the ROC curve (AUC) values were calculated to evaluate the diagnostic performance of the proteins showing q-values less than 0.05 in distinguishing between RRMS patients and OND subjects. The pROC package was used for this purpose; 95% confidence intervals (CI) of AUC were determined by the bootstrap method (2000 replicates used). Binomial and multinomial LASSO regressions (glmnet R package) were used to differentiate among three groups: OND, MS OCB+ and MS OCB−. Cross-validation was performed to identify the optimal penalization parameter (λ) for the LASSO regression models. Student’s t-test was used to check for age matching between the training sets in the multivariate analyses for MS OCB+ vs. OND and MS OCB− vs. OND comparisons.

## 5. Conclusions

In conclusion, we identified a set of six CSF immunological proteins (CD5, IL12B, TNFB, TNFSF14, TNFRSF9 and MIP-1-alpha), each of them having discriminatory power in distinguishing MS from controls. These markers predominantly reflect T cell activation, chronic meningeal inflammation and chemokine-mediated immune responses. Employing a machine learning methodology allowed us to develop diagnostic models that demonstrated satisfactory accuracy. The model designed for MS OCB+ exhibited high specificity and moderate sensitivity, suggesting its potential value in aiding the diagnosis of MS, especially when considered alongside other sensitive diagnostic tools. However, the model for MS OCB− exhibited lower accuracy, indicating potential differences in pathophysiology between MS with and without OCB. The identification of these biomarkers holds promise for enhancing the accuracy of MS diagnosis and offering insights into MS subgroup-specific disease mechanisms. Further validation and testing against relevant clinical comparisons are necessary to establish the robustness and clinical utility of these biomarkers. Additionally, the distinct profiles observed between MS patients with and without OCB raise questions about differing pathophysiological mechanisms and warrant further investigation.

## Figures and Tables

**Figure 1 ijms-25-00139-f001:**
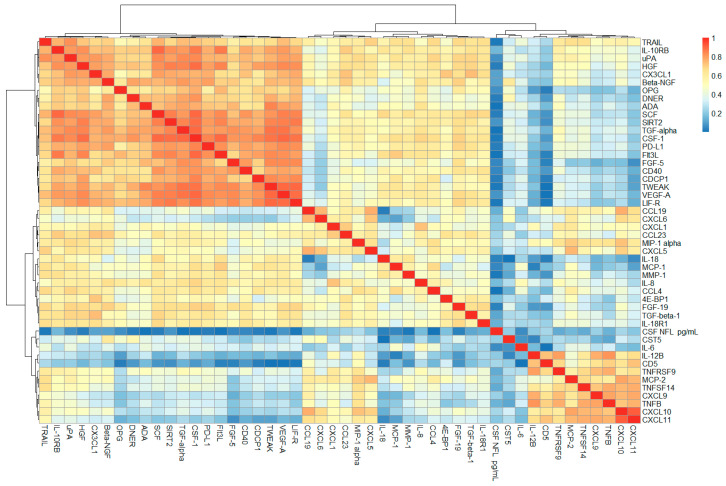
Correlation heatmap. Correlation coefficients were computed according to Spearman and are displayed in absolute values. Two clusters emerge: (i) TRAIL, IL-10RB, uPA, HGF, CX3CL1, Beta-NGF, OPG, DNER, ADA, SCF, SIRT2, TGF-alpha, CSF-1, PD-L1, Flt3L, FGF-5, CD40, CDCP1, TWEAK, VEGF-A, LIF-R; and (ii) IL-12B, CD5, TNFRSF9, CXCL9, TNFB, CXCL10, CXCL11.

**Figure 2 ijms-25-00139-f002:**
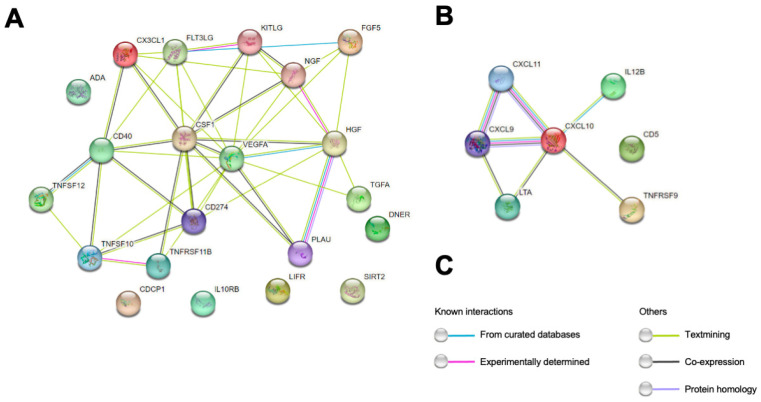
Protein–protein interaction maps of clustered proteins identified in this study. (**A**) First major cluster. (**B**) Second minor cluster. (**C**) Nodes are representative of protein species, and different line colors show the types of evidence for the association. The STRING tool (http://www.string-db.org (accessed on 1 November 2023)) was used to construct the interaction networks. Interactive networks are available at: (**A**) https://version-11-5.string-db.org/cgi/network?networkId=bI2OAa4S6dS1 (accessed on 1 November 2023) (**B**) https://version-11-5.string-db.org/cgi/network?networkId=b5Rvxhh6MPHk (accessed on 1 November 2023).

**Figure 3 ijms-25-00139-f003:**
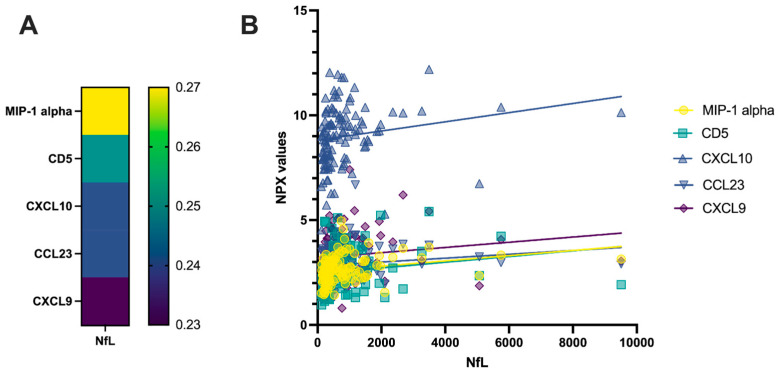
Significant correlations between PEA-tested proteins and CSF NfL in pwMS. (**A**) Correlation coefficients depicted as heatmap (MIP-1-alpha: 0.27, *p* = 0.0067; CD5: 0.25, *p* = 0.012; CXCL10: 0.24, *p* = 0.015; CCL23: 0.24, *p* = 0.018; CXCL9: 0.23, *p* = 0.025). (**B**) Plots showing simple linear regression between PEA-tested proteins and CSF NfL.

**Figure 4 ijms-25-00139-f004:**
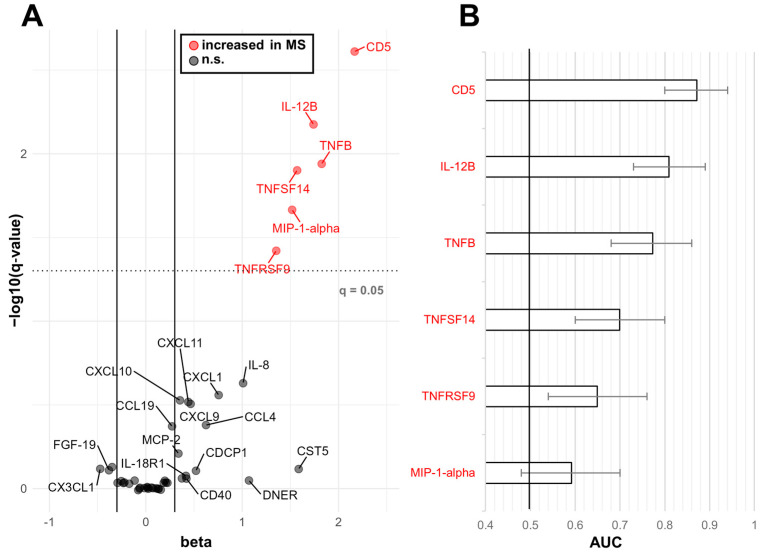
CSF proteins differentially expressed between RRMS patients and OND. (**A**) Logistic regression was applied to determine, for each of the proteins considered, age-, sex- and FDR-adjusted *p*-value (q-value). The coefficients resulting from logistic regressions (beta) are displayed on the *x*-axis while the negative logarithm (base 10) of the q-value is reported on the *y*-axis. (**B**) Area under the ROC curve (AUC) determined for RRMS vs. OND for each of the markers showing a q-value < 0.05 with error bars representing the 95% CI.

**Table 1 ijms-25-00139-t001:** Confusion table showing the performance of the multinomial LASSO model in the training set in the comparison between MS and OND (data are shown for MS OCB+ and MS OCB− separately).

		Predicted Condition		
	Total 118	MS OCB+ 67	MS OCB− 5	OND 46
**Actual** **condition**	MS OCB+ 58	55	10	2
	MS OCB− 24	1	3	1
	OND 36	2	11	33

MS OCB+: multiple sclerosis with cerebrospinal fluid IgG oligoclonal bands. MS OCB−: multiple sclerosis without cerebrospinal fluid IgG oligoclonal bands. OND: other neurological diseases.

**Table 2 ijms-25-00139-t002:** Confusion tables showing the performance of the binomial LASSO model in the training set (A) and in the validation set (B) for the MS OCB+ vs. OND comparison.

A		Predicted Condition		B		Predicted Condition	
	Total 40	MS OCB+ 22	OND 18		Total 54	MS OCB+ 36	OND 18
**Actual condition**	MS OCB+ 22	20	2	**Actual condition**	MS OCB+ 36	29	7
	OND 18	2	16		OND 18	7	11

MS OCB+: multiple sclerosis with cerebrospinal fluid IgG oligoclonal bands. OND: other neurological diseases.

**Table 3 ijms-25-00139-t003:** Confusion tables showing the performance of the binomial LASSO model in the training set (A) and in the validation set (B) for the MS OCB− vs. OND comparison.

A		Predicted Condition		B		Predicted Condition	
	Total 30	MS OCB− 15	OND 15		Total 30	MS OCB− 9	OND 21
**Actual condition**	MS OCB− 15	13	2	**Actual condition**	MS OCB− 9	5	4
	OND 15	2	13		OND 21	4	17

MS OCB−: multiple sclerosis without cerebrospinal fluid IgG oligoclonal bands. OND: other neurological diseases.

**Table 4 ijms-25-00139-t004:** Patient characteristics. Data are expressed as number (percentage), mean ± standard deviation or median (IQR).

	OND	RRMS	*p*-Value *	MS OCB+	MS OCB−	*p*-Value **
N	36	82	-	58	24	-
Sex–F	18 (50%)	58 (70.7%)	<0.05	41 (70.7%)	17 (70.8%)	n.s.
Age (years)	57.9 ± 16.6	37.9 ± 10.6	<0.01	37 ± 10.3	40.2 ± 11.4	n.s.
EDSS	-	1.9 ± 1	-	1.8 ± 0.8	2.3 ± 1.5	n.s.
Disease duration (months)	-	3 (23.5)	-	3 (23.6)	2 (23.5)	n.s.
Recent relapse (<30 days)	-	60 (73.2%)	-	43 (74.1%)	17 (70.8%)	n.s.
CSF NfL (pg/mL)	361 (504)	361 (504)	361 (504)	633 (929)	637.5 (558)	n.s.

* OND vs. RRMS. ** MS OCB+ vs. MS OCB−. CSF: cerebrospinal fluid. EDSS: expanded disability status scale. IQR: interquartile range. MS OCB+: multiple sclerosis with cerebrospinal fluid IgG oligoclonal bands. MS OCB−: multiple sclerosis without cerebrospinal fluid IgG oligoclonal bands. NfL: neurofilament light chain. n.s.: not significant. OND: other neurological diseases. RRMS: relapsing remitting multiple sclerosis.

## Data Availability

The data presented in this study are available on request from the corresponding author.

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
