# Peer review of "The Immune Signature of CSF in Multiple Sclerosis with and without Oligoclonal Bands: A Machine Learning Approach to Proximity Extension Assay Analysis"

_ijms, 2023, doi:10.3390/ijms25010139_

Round 1

Reviewer 1 Report

Comments and Suggestions for Authors

This study aimed to profile immune activation biomarkers in the cerebrospinal fluid (CSF) of individuals with multiple sclerosis (MS) and explore distinct signatures between those with and without oligoclonal bands (OCB). The results identified biomarker panels with high discriminatory power for MS with OCB, but the results for MS without OCB did not exhibit high accuracy. This study may contribute to the understanding of MS biomarkers."

Specific comments.

1. In the abstract, why does the abbreviation of neurological controls is OND?

2. What does the “3” of 3 (23.5) in Table 1 mean?

3. Line 116. Check “ass”.

4. Are there distinct clinical features between MS with OCB and MS without OCB? If MS with OCB exhibits more obvious symptoms than MS without OCB, the diagnosis of MS with OCB could be easier that of MS without OCB. The critical challenge might be discriminating between MS without OCB and controls, rather than between MS with OCB. I think the authors should clarify this point.

5. In conclusion, you conclude that “the distinct profiles observed between MS patients with and without OCB raise questions about differing pathophysiological mechanisms and warrant further investigation.”. It is not clear whether the distinct profiles observed between MS patients with and without OCB are caused by some specific patients or general patients in your sample. Is it possible that only a few patients exhibit significantly different expression levels of proteins, leading to distinct profiles between the two groups?  In the supplementary file, you only display scatter plots between RRMS and OND, omitting plots between OCB+ and OCB-.

Author Response

Reviewer 1

This study aimed to profile immune activation biomarkers in the cerebrospinal fluid (CSF) of individuals with multiple sclerosis (MS) and explore distinct signatures between those with and without oligoclonal bands (OCB). The results identified biomarker panels with high discriminatory power for MS with OCB, but the results for MS without OCB did not exhibit high accuracy. This study may contribute to the understanding of MS biomarkers.

We appreciate the Reviewer's thoughtful assessment of our study, and we thank her/him for the comments to the manuscript.

Specific comments.

  1. In the abstract, why does the abbreviation of neurological controls is OND?

Thank you for having noted this imprecision. Since OND stands for “other neurological diseases”, we have now changed the abstract as follows: “controls with other neurological diseases (OND)”.

  1. What does the “3” of 3 (23.5) in Table 1 mean?

In Table 1, data are presented as number (percentage - %), mean ± standard deviation, or median (interquartile range - IQR). The notation '3 (23.5)' specifically refers to the median disease duration in months, which is 3, with an associated IQR of 23.5. To clarify, the format '3 (23.5)' represents the median (IQR) for disease duration.

  1. Line 116. Check “ass”.

We apologize for the typographical error on line 116 where 'ass' appears instead of 'associated.' Thank you for bringing this to our attention. We have corrected this mistake in the revised manuscript.

  1. Are there distinct clinical features between MS with OCB and MS without OCB? If MS with OCB exhibits more obvious symptoms than MS without OCB, the diagnosis of MS with OCB could be easier that of MS without OCB. The critical challenge might be discriminating between MS without OCB and controls, rather than between MS with OCB. I think the authors should clarify this point.

We appreciate the comment from the Reviewer regarding the distinction between MS with and without OCB and its potential impact on clinical features. We found no significant differences in clinical features between MS patients with and without OCB in our dataset. Of note, as the cohort was retrospectively selected, we limit the comparison to some demographical and clinical data (sex, age, EDSS, disease duration, the presence of a recent relapse and CSF NfL levels), as reported in Table 1.

We agree with the Reviewer that discriminating between MS without OCB and controls could be a critical challenge. In the revised discussion, we have emphasized this point and provided additional context on the diagnostic challenges associated with differentiating MS without OCB from other neurological conditions. Below the changes applied [page 8, lines 358-365]: “On the other hand, the model used for the identification of MS OCB- was found to be less effective. This suggests that MS OCB+ and MS OCB- may differ from a pathophysiological perspective, with MS OCB- showing less definite signs of immune activation in CSF. This underscores the complexity in diagnosing MS when CSF OCB are absent. As an example, in discriminating between MS and white matter lesions associated with migraine and vascular lesions, the absence of CSF OCB is the most robust independent predictor of a non-MS diagnosis [34]. Therefore, the investigation of biomarkers able to identify MS OCB- is highly relevant from a clinical perspective and warrants further research.”

  1. In conclusion, you conclude that “the distinct profiles observed between MS patients with and without OCB raise questions about differing pathophysiological mechanisms and warrant further investigation.”. It is not clear whether the distinct profiles observed between MS patients with and without OCB are caused by some specific patients or general patients in your sample. Is it possible that only a few patients exhibit significantly different expression levels of proteins, leading to distinct profiles between the two groups?  In the supplementary file, you only display scatter plots between RRMS and OND, omitting plots between OCB+ and OCB-.

The Reviewer raises a very interesting point. In our cohort, OCB- patients were not consistently classified, even when employing a trinomial model. Notably, half of the MS OCB- patients did not exhibit distinguishable characteristics from OND, while the other half did not differ from MS OCB+. Hence, these distinctions do not stem from the behaviors of individual patients. To underscore this point, and in response to the Reviewer's feedback, we have modified the supplementary figure to include plots that distinguish OCB+ subjects from OCB- subjects.

Reviewer 2 Report

Comments and Suggestions for Authors

Thank you for allowing me to review the article titled "The immune signature of CSF in multiple sclerosis with and without oligoclonal bands: a machine learning approach to proximity extension assay analysis" (ijms-2727148). This study aims to extensively profile immune activation biomarkers in cerebrospinal fluid among individuals with multiple sclerosis, examining the differing signatures between cases with and without oligoclonal bands. Utilizing advanced machine-learning statistical models, the research seeks to pinpoint specific biomarker combinations that can distinguish people with multiple sclerosis with and without OCB 83 from controls. It also aims to shed light on the disease's pathophysiology.

The abstract is well-structured, but it would benefit from including confidence intervals for the sensitivity and specificity of the applied techniques to enhance the information presented.

The introduction effectively sets the stage for the study's hypothesis and employs pertinent literature on the topic. However, the study's objectives could be articulated more clearly and precisely.

In the materials and methods section, it would be beneficial to detail the study design, including whether a sample size calculation was performed or if the sample was chosen for convenience.

In the results section, adding the confidence intervals for the presented sensitivity and specificity would be valuable to assess the differences among the various techniques applied.

In the discussion, there's no need to restate the study's objective. Instead, it should consider the study's limitations and strengths.

Author Response

Reviewer 2

Thank you for allowing me to review the article titled "The immune signature of CSF in multiple sclerosis with and without oligoclonal bands: a machine learning approach to proximity extension assay analysis" (ijms-2727148). This study aims to extensively profile immune activation biomarkers in cerebrospinal fluid among individuals with multiple sclerosis, examining the differing signatures between cases with and without oligoclonal bands. Utilizing advanced machine-learning statistical models, the research seeks to pinpoint specific biomarker combinations that can distinguish people with multiple sclerosis with and without OCB from controls. It also aims to shed light on the disease's pathophysiology.

We sincerely appreciate the Reviewer's positive acknowledgment of our article.

The abstract is well-structured, but it would benefit from including confidence intervals for the sensitivity and specificity of the applied techniques to enhance the information presented.

Thank you for your comment. In response, we recognize the value of confidence intervals for assessing the precision of estimates. However, in our study with separate training and validation cohorts, adding confidence intervals may not offer meaningful insights. Splitting the dataset into these subsets limits our ability to generate robust confidence intervals due to the small validation sample size. While confidence intervals are helpful for indicating model reliability, given our sample size constraints, we find a more comprehensive evaluation of model performance in a small validation set preferable. We chose not to include confidence intervals in the manuscript to prevent potential confusion among readers about generating reliable intervals in our study design.

For transparency, we present the confidence intervals here as follows:

Model for MS OCB+: Sensitivity: 91% [90-100%] in training, 81% [0.6-100%] in validation. Specificity: 94% [78-100%] in training, 94% [75-100%] in validation.

Model for MS OCB-: Sensitivity: 87% [60-100%] in training, 56% [33-100%] in validation. Specificity: 80% [67-100%] in training, 48% [24-100%] in validation.

The introduction effectively sets the stage for the study's hypothesis and employs pertinent literature on the topic. However, the study's objectives could be articulated more clearly and precisely.

Thank you for this constructive comment regarding the clarity and precision of our study objectives. In response to your feedback, we have carefully revisited and refined the articulation of our study objectives in the introduction to ensure they are presented more clearly and precisely.

Below, the changes applied [page 2, lines 81-88]: “In the present study, we utilized PEA technology to comprehensively measure a panel of biomarkers reflecting immune activation in CSF samples obtained from pwMS with and without OCB and a control group. Additionally, we quantified CSF levels of neurofilament light chain (NfL), a well-established biomarker of axonal damage [13], in order to identify the prevailing immunological profile more closely linked with axonal injury. By applying robust machine-learning statistical models, the aim of the study was to identify combinations of biomarkers able to differentiate pwMS with and without OCB from controls, as well as to provide novel insights into the pathophysiology of the disease.”

In the materials and methods section, it would be beneficial to detail the study design, including whether a sample size calculation was performed or if the sample was chosen for convenience.

We thank the Reviewer for highlighting this aspect. In the manuscript, particularly in the materials and methods section, we have explicitly mentioned that patient enrollment occurred consecutively between 2014 and 2017. We also specified that we selected patients between those who underwent lumbar puncture in our Neurology Clinic for diagnostic purposes within this time frame. Therefore, we included patients diagnosed with multiple sclerosis and those with other neurological diseases (OND), representing minor neurological conditions and serving as our control group. The retrospective design of the study might represent a limitation of our manuscript. We therefore decided to include a section at the end of the discussion where we now report the strengths and limitations of the study. Specifically, we now state [page 9, lines 421-424]: “In conclusion, one potential limitation of our study is its retrospective design. The inclusion of participants was based on the availability of stored CSF samples over a three-year period, which may not constitute a random selection. However, we believe this does not introduce selection bias to our cohort.”

In the results section, adding the confidence intervals for the presented sensitivity and specificity would be valuable to assess the differences among the various techniques applied.

Thank you for this point. Please, see above the answer to a comment of confidence intervals to be included in the abstract.

In the discussion, there's no need to restate the study's objective. Instead, it should consider the study's limitations and strengths.

Thank you for this point. We have now removed the first part of the discussion, which has been moved to the introduction of the manuscript in order to better clarify there the study’s objective. At the end of the discussion, we have now included a summary of the strengths and limitations of the study as it follows [page 9-10, lines 421-434]: “In conclusion, one potential limitation of our study is its retrospective design. The inclusion of participants was based on the availability of stored CSF samples over a three-year period, which may not constitute a random selection. However, we believe this does not introduce selection bias to our cohort. Our research has several strengths: i) we addressed the clinical challenge of diagnosing MS by focusing on pwMS with and without OCB, a subset that poses diagnostic difficulties [8]. This reflects the study's relevance to real-world diagnostic scenarios; ii) we employed an innovative technology. The study utilized PEA technology, specifically the PEA technology, which offers a promising tool for quantitatively measuring protein levels in various body fluids, including CSF. This innovative approach enhances the precision of biomarker exploration; iii) to interpret the complexity of data generated by PEA, we applied a machine-learning statistical approach that enhances the robustness of our findings [42], and we built training and validation sets to test the accuracy of biomarkers in discriminating between group.”

Reviewer 3 Report

Comments and Suggestions for Authors

- Introduction: At the end of the section add clearly stated scientific contributions of the paper. 

- Results are mixed up with data analysis and patients information. It should be reorganized. Section 4 should be before Results. Parts of the current Result section should be in Materials and Methods.

- Figures 1 and 3 should be explained int he text. Also what are the heatmaps and how to interpret them.

- Figure 1 B - link doesn't work.

- You should add references from 2023. Also try to incorporate them tot the context of machine learning with the application in MS.

- Overall assessment: the paper should be reorganized and resubmitted to review.

Author Response

Reviewer 3

Introduction: At the end of the section add clearly stated scientific contributions of the paper. 

We thank the Reviewer for her/his suggestion. We have now modified the final part of the Introduction, where we have included the following statement [page 2, lines 81-88]: “In the present study, we utilized PEA technology to comprehensively measure a panel of biomarkers reflecting immune activation in CSF samples obtained from pwMS with and without OCB and a control group. Additionally, we quantified CSF levels of neurofilament light chain (NfL), a well-established biomarker of axonal damage [13], in order to identify the prevailing immunological profile more closely linked with axonal injury. By applying robust machine-learning statistical models, the aim of the study was to identify combinations of biomarkers able to differentiate pwMS with and without OCB from controls, as well as to provide novel insights into the pathophysiology of the disease.”

Results are mixed up with data analysis and patients information. It should be reorganized. Section 4 should be before Results. Parts of the current Result section should be in Materials and Methods.

In response to the Reviewer's recommendation, we have relocated the paragraph detailing the characteristics of the patients from the Results section to the Materials and Methods section. In the manuscript we have reported the Methods (section 4) after the Results and the Discussion, because this is required by the editorial setting of the paper. In submitting the revised version of the manuscript, we will ask the editors for the opportunity to change the structure of the manuscript with the Materials and Methods before the results and Discussion.

Figures 1 and 3 should be explained int he text. Also, what are the heatmaps and how to interpret them.

Thank you for this comment. We have explained figure 1 and 3 in the Results section (subsection 2.1). Specifically, we explained that we found two clusters of proteins correlated each other (heatmap, figure 1) and a group of proteins positively correlated with CSF NfL (figure 3). Further, in the discussion these findings are contextualized in detail.  

Figure 1 B - link doesn't work.

We apologize for this point. We have now generated a new permanent link from the web site of the String tool, which currently works.

You should add references from 2023. Also try to incorporate them tot the context of machine learning with the application in MS.

We have now included the following references in the text:

Jakimovski D, Bittner S, Zivadinov R, Morrow SA, Benedict RH, Zipp F, Weinstock-Guttman B. Multiple sclerosis. Lancet. 2023 Nov 7:S0140-6736(23)01473-3. doi: 10.1016/S0140-6736(23)01473-3. Epub ahead of print. PMID: 37949093.

Solomon AJ, Arrambide G, Brownlee WJ, Flanagan EP, Amato MP, Amezcua L, Banwell BL, Barkhof F, Corboy JR, Correale J, Fujihara K, Graves J, Harnegie MP, Hemmer B, Lechner-Scott J, Marrie RA, Newsome SD, Rocca MA, Royal W 3rd, Waubant EL, Yamout B, Cohen JA. Differential diagnosis of suspected multiple sclerosis: an updated consensus approach. Lancet Neurol. 2023 Aug;22(8):750-768. doi: 10.1016/S1474-4422(23)00148-5. PMID: 37479377.

Katsarogiannis E, Landtblom AM, Kristoffersson A, Wikström J, Semnic R, Berntsson SG. Absence of Oligoclonal Bands in Multiple Sclerosis: A Call for Differential Diagnosis. J Clin Med. 2023 Jul 13;12(14):4656. doi: 10.3390/jcm12144656. PMID: 37510771; PMCID: PMC10380970.

Hossain MZ, Daskalaki E, Brüstle A, Desborough J, Lueck CJ, Suominen H. The role of machine learning in developing non-magnetic resonance imaging based biomarkers for multiple sclerosis: a systematic review. BMC Med Inform Decis Mak. 2022 Sep 15;22(1):242. doi: 10.1186/s12911-022-01985-5. PMID: 36109726; PMCID: PMC9476596.

Overall assessment: the paper should be reorganized and resubmitted to review.

We thank the Reviewer for her/his constructive comments and suggestions. We believe that the quality of our work is now improved.

Round 2

Reviewer 1 Report

Comments and Suggestions for Authors

The authors have addressed most of my comments.

Reviewer 3 Report

Comments and Suggestions for Authors

I'm not satisfied with elaboration of scientific contribution. Although, it is interesting study, and results. What is new in science?